# CoMotion: Concurrent Multi-person 3D Motion

**Alejandro Newell**    **Peiyun Hu**    **Lahav Lipson**    **Stephan R. Richter**    **Vladlen Koltun**

Apple

## ABSTRACT

We introduce an approach for detecting and tracking detailed 3D poses of multiple people from a single monocular camera stream. Our system maintains temporally coherent predictions in crowded scenes filled with difficult poses and occlusions. Our model performs both strong per-frame detection and a learned pose update to track people from frame to frame. Rather than match detections across time, poses are updated directly from a new input image, which enables online tracking through occlusion. We train on numerous image and video datasets leveraging pseudo-labeled annotations to produce a model that matches state-of-the-art systems in 3D pose estimation accuracy while being faster and more accurate in tracking multiple people through time.

## 1 INTRODUCTION

Some of the most exciting applications of computing require understanding where people are and how they move. A single monocular camera can support a variety of human-centric applications if it is coupled with a system that can model the motion of all humans in its visual field. For the broadest range of applications, we are interested in systems that (a) see multiple people, (b) estimate their poses in 3D, (c) track their poses through time in a monocular video stream, even through occlusion, and (d) do so online, in a streaming fashion, processing each frame and updating everyone's estimated 3D poses without peeking into the future.

For such a system to work in the wild, it must handle cluttered scenes that are crowded with people. As people move about, the model must keep track of who is who from frame to frame in order to produce a coherent 3D motion estimate. This can be difficult as people occlude each other, pass behind objects, and step out of view. Maintaining accurate 3D pose estimates online is particularly challenging, as the model cannot leverage future context to fill in the gap for a missing observation. Instead, the model must forecast to the best of its ability and snap to the correct state as soon as the person is visible again.

Many existing approaches to pose tracking follow a two-stage detect-and-associate paradigm (Insafutdinov et al., 2017; Xiao et al., 2018; Doering et al., 2022; Rajasegaran et al., 2022). The idea is to run a state-of-the-art pose estimation system for each frame, and then link poses across frames using cues such as proximity and appearance. As with many two-stage approaches, the second-stage pose estimates and tracks can be undermined by flaws in the first stage. If a detection is missed in the first stage, some other mechanism must be introduced to update the pose. For example, one option is to fill in any missing poses offline using the context of the full track (Goel et al., 2023). However, this solution is not suitable for online applications that deal with live video.

We present CoMotion, a video-based approach that performs frame-to-frame pose updates in a fundamentally different manner. Following the *tracking by attention* paradigm (Meinhardt et al., 2022), we do not link independent per-frame detections – rather, we train a recurrent model that maintains a set of tracked 3D poses and updates them when a new frame arrives. To update the poses, CoMotion directly ingests the pixels of the new frame. The model uses whatever image evidence is available to update the poses of all people in the scene simultaneously – even those who may not be currently visible. One advantage of this approach is that CoMotion can learn to utilize subtle pixel cues. A pair of feet poking out may be insufficient to trigger an independent detection, but can still be quite informative if the model has been following that person's movement.

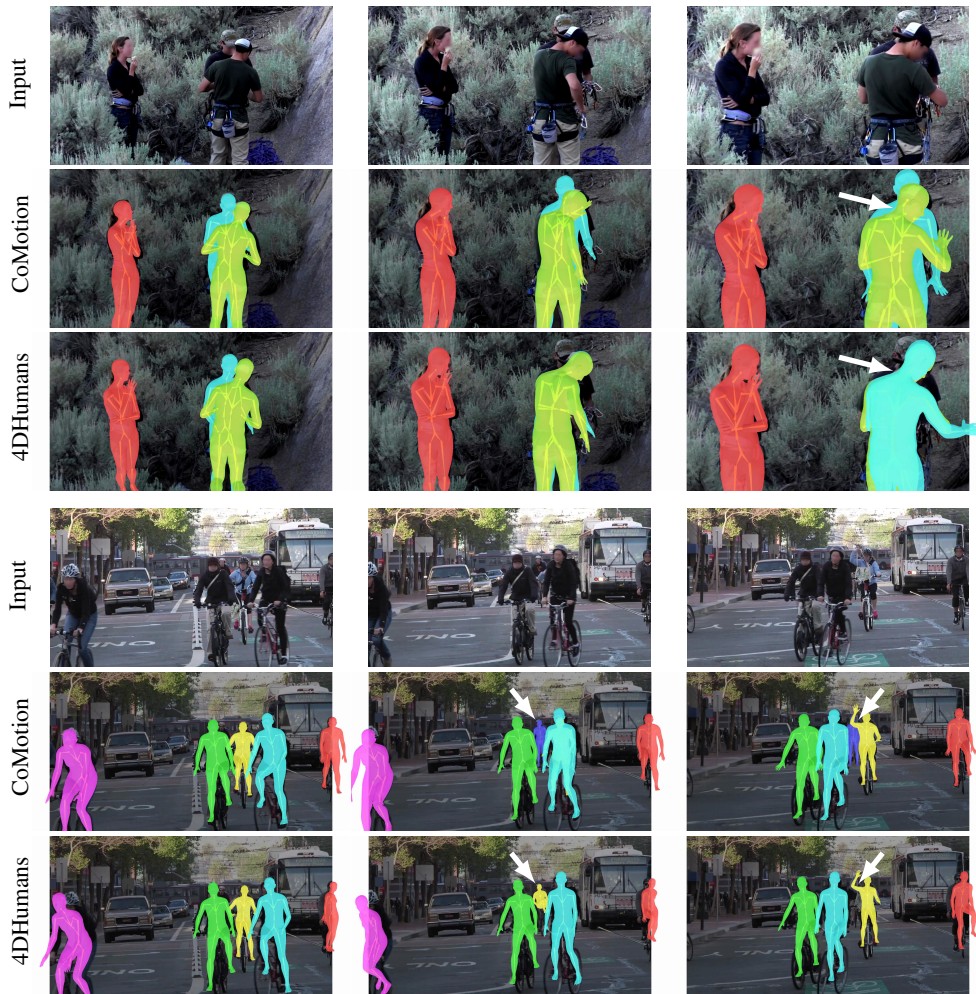

Figure 1: CoMotion tracks 3D poses online from monocular RGB video. Rather than detect new poses in each frame and associate them to existing tracks, CoMotion updates tracks directly from incoming image features. As a result, CoMotion keeps track of distinct individuals as they overlap in the camera frame (top) and occlude each other (bottom). Arrows highlight some points of interest.

Data is a challenge with such a video-based approach, for both training and evaluation. No dataset exists that provides ground-truth 3D poses of groups of people moving in diverse, cluttered real-world scenes. As a result, it is rare to find video-based 3D multi-person pose methods that support in-the-wild videos with large numbers of people. Comparable methods evaluate tracking on limited settings with a handful of people (Reddy et al., 2021; Sun et al., 2023). Only top-down methods report tracking results on more challenging real-world videos (Rajasegaran et al., 2022).

We find that we can achieve excellent performance by training on a heterogeneous mixture of datasets. Each dataset offers a complementary form of supervision – some are single-image, some are video, some are synthetic, and some provide partial ground truth. In particular, we leverage pseudo-labeling to supervise our system on challenging real-world video (Andriluka et al., 2018; Sun et al., 2022) that would be exceedingly difficult to annotate otherwise. We are unaware of any prior work in this area that trains on a comparable heterogenous mix of datasets. In particular, there are no multi-person 3D pose methods that supervise temporally unrolled predictions on complex videos such as those provided by PoseTrack (Andriluka et al., 2018; Doering et al., 2022).

We evaluate pose estimation performance and multi-person tracking using standard datasets for each task. CoMotion exhibits state-of-the-art pose estimation performance, while providing better and faster tracking with fewer temporal artifacts such as jitter and dropped poses. Figure 1 illustrates this

combination of 3D pose estimation accuracy and tracking stability. On PoseTrack21 (Doering et al., 2022), CoMotion improves MOTA by 14% and IDF1 by 12% relative to the prior state of the art. At the same time, CoMotion is an order of magnitude faster than the prior state of the art.

## 2 RELATED WORK

Many approaches to pose estimation and tracking have been explored. Some detect poses only in a single image, some operate on video but only handle a single person, and some track multiple people in video but don't estimate their poses. Few take on the holistic setting of predicting and tracking the poses of multiple people in video, let alone in a streaming (online) manner. For an accessible overview of the different perspectives, we classify relevant work along two dimensions, namely the input modality (single image or video) and the handling of multiple people.

**Single-person, single-frame 3D pose.** Many approaches to 3D human pose estimation predict a pose from a single image that is cropped to the target person (Bogo et al., 2016; Kanazawa et al., 2018; Pavlakos et al., 2018). Poses are commonly represented by the SMPL model (Loper et al., 2015), which defines a mesh from joint angles and shape parameters. The SMPL parameters can be regressed directly from an input image (Goel et al., 2023). Another common strategy is to employ a feedback loop that alternates between predicting SMPL parameters and observing the reprojection to image space to inform further updates (Zhang et al., 2021a; 2023a;b;c; Wang & Daniilidis, 2023). Much progress in this setting can be attributed to scaling up datasets, model sizes, and training times (Goel et al., 2023; Khirodkar et al., 2024; Sárándi & Pons-Moll, 2024).

**Single-person pose from video.** This setting assumes that a person has already been detected and tracked, such that it can be presented to the model through a sequence of cropped frames or bounding boxes. A line of work operates on pre-extracted 2D keypoints (Zheng et al., 2021; Zhang et al., 2022; Tang et al., 2023; Zhao et al., 2023) with transformers because the sparse input signal enables modeling of large temporal windows. Alternatively, recurrent networks enable online aggregation of features between frames. VIBE (Kocabas et al., 2020) and PMCE (You et al., 2023) use a GRU, and RoHM (Zhang et al., 2024) uses a denoising diffusion model to decode subsequent frames. Our approach also uses a GRU to run inference on video. However, where these prior methods rely on an external tracking solution and can only perform inference on a single subject, our method does its own tracking to handle multiple subjects in parallel.

**Multi-person, single-frame pose.** A straightforward approach to predicting poses of multiple people is detecting their individual bounding boxes and processing those via a single-person pose estimator (Goel et al., 2023; Fang et al., 2017; Papandreou et al., 2017; Huang et al., 2017). However, this approach scales poorly to crowded scenes. More recently, several works extract a dense feature map from the input image and perform cross-attention on a set of query tokens regressed from a ResNet or ViT applied to the same image (Liu et al., 2023; Shi et al., 2022; Baradel et al., 2024). In these approaches, the costly dense feature extractor is only run once per image. We also apply cross-attention, but our model produces queries in each timestep conditioned on past history and must also attend to people who may not be visible in the current frame (e.g., because they temporarily move out of the camera's field of view), which would not be relevant in the single-image setting.

**Multi-person pose from video.** A key challenge in multi-person pose from video is associating new observations with pre-existing tracks. An early approach was to estimate a graph of pose keypoints with intra-/inter-frame edge weights and then estimate connected components by solving a minimum-cost multi-cut problem (Insafutdinov et al., 2017). Better results were later obtained using a track-and-detect paradigm, where independent per-frame detections are grouped into tracks using visual keypoint features and image location (Girdhar et al., 2018) as well as 2D motion (Xiao et al., 2018). More recently, PHALP (Rajasegaran et al., 2022) adopted this mechanism for 3D pose rather than 2D, and also used the predicted 3D poses as an additional feature for grouping. 4D Humans (Goel et al., 2023) went further with a more general-purpose pose representation (Loper et al., 2015) and new architecture. These approaches achieve excellent results on modern benchmarks but scale poorly on videos with many subjects, since they run a pose estimator independently on each cropped detection. In contrast to these tracking-by-detection methods, CoMotion estimates poses and localizes existing tracks in parallel, reasoning over all poses simultaneously. In addition to a more holistic treatment of the scene, this approach is an order of magnitude faster.

Figure 2: **Overview.** CoMotion estimates 3D poses for all people in a frame. An image encoder produces image features $F^t$, which are passed through the detection module to identify potential new tracks. In parallel, the pose update module attends to $F^t$ to update the existing tracks from the previous timestep. Both outputs are compared to each other to decide whether to instantiate or remove any tracks. If a detection is flagged as a new track, it is passed through the update module before being added to the final output tracks for the current frame. The inset details the pose update module.

Unlike tracking-by-detection, tracking-by-attention (Meinhardt et al., 2022; Sun et al., 2020) detects new objects and updates existing tracks jointly by performing cross-attention between an image and a set of learned per-track and new-object query tokens. While many of these works condition the query tokens only on the previous few frames (Zeng et al., 2022), MeMOTR maintains a long-term memory of per-track query tokens and gradually updates them based on new observations (Gao & Wang, 2023). We follow the tracking-by-attention paradigm in this work, but find that there are many challenges to instantiating this paradigm for multi-person 3D pose tracking, including the nature of available training data. Bounding-box-based methods can get away with training on pairs of frames (Meinhardt et al., 2022), but we find that it is critical to unroll on longer video sequences during training to learn to track detailed 3D poses accurately.

## 3 CoMotion

**Preliminaries.** Given a sequence of monocular RGB images $\{I^1, I^2, ..., I^t\} \in \mathbb{R}^{h \times w \times 3}$, we aim to produce 3D pose estimates for each person in the scene at each time-step $t$ conditioned only on the past and the present (no peeking into the future). Each prediction should be associated with a particular identity such that we know all poses belonging to a given individual. Predictions are provided as complete trajectories from the beginning to the end of a track. Our system does not model tracks through long occlusions nor does it perform re-identification to link tracks.

We follow standard practice (Bogo et al., 2016; Kanazawa et al., 2018) and parameterize each person with SMPL (Loper et al., 2015), which consists of a translation term $\gamma \in \mathbb{R}^3$, joint angles $\theta \in \mathbb{R}^{72}$ (including a global orientation), and shape parameters $\beta \in \mathbb{R}^{10}$.

All estimates are made in the camera coordinate frame. We do not explicitly model any changes to the camera pose or intrinsics (e.g., due to camera motion or zooming). During inference, the system accepts as input an intrinsics matrix $K$. All output 3D estimates will project correctly back into the image according to this provided matrix using the pinhole camera model. If no ground truth intrinsics are available, we fall back to reasonable defaults. To increase robustness against incorrect intrinsics, we augment $K$ during training allowing us to make predictions on in-the-wild videos by providing either a generic default setting for $K$ or using the correct intrinsics when they are available.

**Overview.** CoMotion is a complete tracking stack which consists of logic for starting and ending tracks, and a means to update tracks from frame to frame. Figure 2 provides an overview of the architecture and logic. Given the current frame, the *detection module* detects 3D poses to serve as candidates for new tracks. The *pose update module* adjusts the poses and hidden states of all existing tracks. Both modules operate directly on image features $F^t$ produced by an image encoder – in our case a standard ConvNextV2 model (Woo et al., 2023). We compare detections to existing tracks in order to determine if any new tracks should be created and to identify which (if any) existing tracks are stale. We describe each component in the following paragraphs, and refer to the appendix for additional details.

**Detection module.** Our detection module is single-shot (Liu et al., 2016) and consists of a detection head that applies several consecutive downsampling and ConvNeXt blocks (Woo et al., 2023) to the given image features $F^t$, yielding a low-resolution feature pyramid. At each level of the pyramid, a 1x1 convolution decodes candidate detections. Instead of multiple anchor boxes as in a traditional bounding-box approach, our detector produces multiple candidate poses at each spatial location, each consisting of SMPL parameters and a confidence value. After pruning candidate poses via non-maximum suppression, we complement poses with hidden states to form detected tracks for the current frame. More formally, we define a track $X = (\gamma, \theta, \beta, h)$ for each detection, where $h$ is a zero-initialized hidden state vector.

**Pose update module.** The pose update module refines a set of tracks given image features $F^t$. Let $\mathbf{X}^t = \bigcup_{i=1}^{N} X_i^t$ be the union of all existing tracks $X_i^t$ for persons $i$ at timestep $t$. We can then write the update to poses and hidden states of existing tracks as $\mathbf{X}^t = \text{update}(F^t, \mathbf{X}^{t-1})$.

The inset on the right of Fig. 2 illustrates the high-level architecture of the update module. We start by encoding $\mathbf{X}^{t-1}$ into a set of tokens. Specifically, from each track's SMPL parameters, we compute 2D projected keypoint locations, then pass the raw SMPL parameters, the 2D keypoints, and the current hidden states through an MLP to produce initial track tokens. Given these tokens, we use cross-attention to query image features $F^t$ attending to the entire image for information about how each pose has changed from the previous timestep. We perform further processing with several transformer layers, from which we decode a new SMPL pose and updated hidden state for each track. We use a GRU for the recurrent hidden state update. For more architecture details, please refer to Appendix A.3.

The model must learn which information in the current frame is relevant for updating each track. This allows refining poses based on partial or indirect observations that may not be enough to trigger a detection. Furthermore, if no visual information is available that pertains to a given track (i.e., during occlusion), the model is still responsible for updating the pose and position of the person.

**Track management.** We use a simple set of heuristics to determine when to delete or instantiate tracks. To guide these heuristics, we measure the similarity between tracks with a modified version of Object Keypoint Similarity (OKS) from COCO (Lin et al., 2014). OKS compares the distance between pairs of keypoints and returns a value between 0 and 1, with 1 indicating perfectly overlapping poses. This is a useful signal as it distinguishes cases where a coarser similarity signal like bounding box intersection-over-union (IOU) would be insufficient. Specifically, we can differentiate two people who occupy the same space in the image but have differing poses (such as a pair of dancers clasped together). For details on our modifications to the OKS calculation see Appendix A.3.4.

*Instantiation.* From all detections $\tilde{\mathbf{X}}^t = \bigcup_{j=1}^{M} \tilde{X}_j^t$, we identify new tracks as those with insufficient similarity to existing tracks (i.e., with $\max_i(\text{OKS}(\tilde{X}_j^t, \mathbf{X}^t)) < 0.2$). We initialize these tracks by refining them via the pose update module which makes slight adjustments to the pose and yields a nonzero initial hidden state. The resulting new tracks are then added to $\mathbf{X}^t$.

*Deletion.* For each track $X_i$, we maintain an exponential moving average of matched OKS scores $s_i^t = \alpha \cdot \max_j(\text{OKS}(\tilde{\mathbf{X}}^t, X_i^t)) + (1 - \alpha) \cdot s_i^{t-1}$ with $\alpha = 0.2$. If this average drops below 0.15, we delete the track. We also delete any track that has existed for fewer than 4 frames and registers an OKS below 0.15 (without the moving average) as this often indicates a spurious detection.

Another important check is whether two tracks overlap for too long. Specifically, if a pair of tracks $(X_i, X_j)$ maintains an OKS above 0.6 for more than 20 consecutive frames (two-thirds of a second), we delete the track with worse alignment to recent detections as indicated by a lower $s^t$. While this heuristic is effective in flagging tracks that have collapsed to the same pose, it may on occasion prematurely end occluded tracks that the network is handling correctly.

Overall, these heuristics serve as simple, quick checks for managing the discrete set of tracks $\mathbf{X}^t$ at no additional computational cost while letting the network do the more difficult task of modeling how people move across time. The specific thresholds and values provided serve to provide a rough sense of how we use the system in practice with reasonable defaults, but these are not magic numbers which need to be adhered to perfectly. A nice feature of CoMotion is that these values can be easily tuned to target different downstream settings, e.g., in some cases it may be helpful to be more forgiving before eliminating uncertain or poorly matched tracks.

## 4 TRAINING

### 4.1 DATASETS

We follow 4D Humans and train on a mixture of diverse pseudo-labeled image datasets to increase robustness on in-the-wild images (Goel et al., 2023). Specifically, we train on InstaVariety (Kanazawa et al., 2019), COCO (Lin et al., 2014), and MPII (Andriluka et al., 2014). As we found the original pseudo-labels provided by 4D Humans to be too sparse for training our detection module, we relabeled the datasets with NLF, a recent state-of-the-art single-person 3D pose model (Sárándi & Pons-Moll, 2024). Further discussion of this relabeling process can be found in Appendix A.4.

In addition, we train on videos with ground truth tracks from PoseTrack (Andriluka et al., 2018) and DanceTrack (Sun et al., 2022). PoseTrack only provides 2D keypoint annotations while DanceTrack only offers bounding boxes, so we pseudo-label the videos with NLF. While this yields 3D annotations for challenging real-world video sequences that would be impossible to obtain otherwise, we note one drawback. The predicted labels are unreliable for people in close proximity, so we flag and ignore these samples during training.

We further train on a large set of sequences with perfect synthetic 3D ground truth. Specifically, we include BEDLAM (Black et al., 2023), which consists of scenes with many people with sampled motions sourced from AMASS (Mahmood et al., 2019), and WHAC-A-MOLE (Yin et al., 2024), which features clips of dancing couples.

### 4.2 CURRICULUM

We follow a three-stage curriculum to train CoMotion.

**Stage 1** pretrains the image encoder and detection module on large batches of single images, namely the pseudo-labeled InstaVariety, COCO, and MPII datasets and BEDLAM. We match candidate detections to ground-truth annotations and supervise the output SMPL terms and detection confidences.

To optimize the model's SMPL predictions, we employ the following loss functions: an $\mathcal{L}_1$-loss to minimize 2D projection error, an $\mathcal{L}_1$-loss for the root-normalized 3D joint position error, and an $\mathcal{L}_2$-loss for the difference in SMPL joint angles. We supervise the betas on samples from BEDLAM with an $\mathcal{L}_1$-loss, but refrain from applying a loss on the betas from the pseudo-labeled samples.

We apply a binary cross-entropy loss to supervise the output confidence term sampling a random subset of unmatched detections to serve as negatives. Additionally, we use a keypoint heatmap loss (Cao et al., 2017) as an additional auxiliary training signal. The model is trained for 400K iterations on 32 A100 GPUs taking approximately 3 days. By the end of this stage, we have a strong single frame multi-person 3D pose estimation system.

**Stage 2** freezes the image encoder and detection module and exclusively trains the pose update module on multiple frames. Our training procedure is simpler than the tracking setting deployed at test time as we do not actively manage tracks during training. That is, at no point do we add or remove tracks as we unroll through a clip. This is a practical detail to support training on minibatches. Instead, on the first frame we match detections to a subset of ground-truth annotations and unroll those specific samples through time.

In this stage, we train on short, 8-frame video clips from BEDLAM and WHAC-A-MOLE as well as the pseudo-labeled PoseTrack and DanceTrack data. We also synthesize videos by panning and zooming on single images from InstaVariety. We train for 200k iterations, which take 3 more days. We supervise the updated poses at each timestep applying the same SMPL losses described in Stage 1.

**Stage 3** extends training of the pose update module to longer video sequences. We sample sequences of length 96 from DanceTrack and length 32 from WHAC-A-MOLE. Samples from such long sequences are particularly slow to train on so we include a mix of shorter 8-frame clips from PoseTrack and images from InstaVariety. To reduce GPU memory consumption, we enable gradient checkpointing. The model is fine-tuned for 50K iterations over 1.5 days.

Overall, this training curriculum allows us to efficiently train a system to handle video by front-loading the initial visual feature learning on single images before the slower video training stage.

Table 1: **Tracking evaluation.** Performance of CoMotion and several baselines on PoseTrack21. We report results using the official evaluation code (top) as well as results after fixing a bug in the evaluation code that caused the 'ignore' regions to be handled incorrectly (bottom, marked with a †).

| Method | PoseTrack21 | | | |
| --- | --- | --- | --- | --- |
| | MOTA↑ | IDF1↑ | IDP↑ | IDR↑ |
| TRMOT (Wang et al., 2020) | 47.2 | 57.3 | 70.0 | 46.6 |
| FairMOT (Zhang et al., 2021b) | 56.3 | 63.2 | 81.0 | 51.8 |
| CorrTrack + ReID (Doering et al., 2022) | 52.0 | 66.5 | 72.4 | 61.4 |
| Tracktor++ (Bergmann et al., 2019) | 59.5 | 69.3 | 76.4 | 63.5 |
| CoMotion (ours) | 67.6 | 77.9 | 83.4 | 73.0 |
| 4DHumans (Goel et al., 2023) † | 56.7 | 70.9 | 87.1 | 59.7 |
| CoMotion (ours) † | 71.4 | 79.5 | 87.1 | 73.0 |

Also, we can fit much larger batches during video training since we do not have to calculate gradients for or update the weights of the image encoder. And even though most training is on very short clips, the model behaves well when unrolled over long temporal sequences spanning minutes.

## 5 EXPERIMENTS

No standard benchmark evaluates the full combination of characteristics that CoMotion was designed for: tracking 3D poses of multiple people through video. For quantitative comparisons to prior work, we evaluate CoMotion on a subset of its capabilities at a time, "projecting" it to existing benchmarks that focus either on tracking stability (without evaluating 3D pose accuracy) or pose estimation on images (without evaluating tracking). We use established benchmarks for 2D and 3D pose estimation and, separately, tracking, and adhere to established evaluation protocols and metrics.

**Tracking.** To evaluate tracking across crowded sequences with interesting poses, we turn to Pose-Track21 (Doering et al., 2022). We use the evaluation code provided by Doering et al. (2022) to compute standard tracking metrics. We report Multi-Object Tracking Accuracy (MOTA), an aggregate score that penalizes missed detections, false positives, and ID switches (100 is a perfect score). Other metrics such as IDF1 and ID precision and recall (IDP and IDR) quantify how well the network preserves a single tracked identity per person. The results are listed in Table 1(top).

We run our full stack on a target input video from beginning to end. Frames are padded and resized to 512x512, and we unroll frames at this fixed resolution (at no point do we perform any cropping to individuals). CoMotion substantially outperforms prior work, improving MOTA by 14% and IDF1 by 12%.

In the course of performing this evaluation and analyzing the results, we found a bug in the Pose-Track21 evaluation code. PoseTrack21 provides partial annotations accompanied by 'ignore' regions that mark parts of images in which ground-truth data is missing. The 'ignore' regions are crucial for evaluation because they signify that tracks predicted in these regions should not be regarded as false positives – they may be correct, but may not align with ground-truth annotations because annotations in these regions are known to be missing. The bug pertains to the handling of these 'ignore' regions and caused many detections in the 'ignore' regions to be incorrectly regarded as false positives. Table 1(bottom) reports results after this bug is fixed. (Here we can only benchmark CoMotion and a method that is reproduced in our environment, since it's not possible to carry numbers over from prior literature.) We will release our fix to PoseTrack21 and recommend that future work adopt the corrected evaluation code. More details can be found in Appendix A.1.

The most competitive baseline that also performs high-quality 3D pose estimation is 4D Humans (Goel et al., 2023). The paper of Goel et al. (2023) reports results on PoseTrack18 (Andriluka et al., 2018) but not on the more accurately and completely annotated PoseTrack21. Unfortunately, due to the drastically incomplete annotations in PoseTrack18 (which motivated the creation of PoseTrack21), we observe that tracking evaluation on PoseTrack18 can be misleading. Many sequences in PoseTrack18 only have annotations for a fraction of the people in the scene, and models are penalized for *correctly* detecting and tracking people who lack annotations. See Sec. A.1 for examples.

Table 2: **PoseTrack18 vs. PoseTrack21.** The annotations in PoseTrack18 are drastically incomplete, penalizing methods that correctly detect and track people in the scene. Indeed, this inspired the creation of PoseTrack21 (Doering et al., 2022), which provides more complete annotations and was released as a direct replacement of PoseTrack18 (same images, more complete annotations). We provide results on PoseTrack18 for backward compatibility with Goel et al. (2023), but strongly recommend that all future work adopt PoseTrack21 instead. (* We rerun the authors' code in order to report performance on PoseTrack21.)

| Method | PoseTrack18 | | | PoseTrack21 | | | | | | |
|---|---|---|---|---|---|---|---|---|---|---|
| | HOTA↑ | IDs↓ | MOTA↑ | MOTA↑ | IDF1↑ | IDP↑ | IDR↑ | FP↓ | FN↓ | FPS↑ |
| 4DHumans (Goel et al., 2023) | 57.8 | 382 | 61.4 | – | – | – | – | – | – | - |
| 4DHumans (reproduced)* | 58.0 | 349 | **61.8** | 56.7 | 70.9 | 87.1 | 59.7 | 7817 | 50652 | 0.51 |
| CoMotion *strict* | **58.2** | **232** | 59.9 | 61.8 | 74.0 | **89.1** | 63.3 | **6086** | 45664 | - |
| CoMotion | 54.9 | 344 | 51.3 | **71.4** | **79.5** | 87.1 | **73.0** | 8115 | **30394** | **5.68** |

Since 4D Humans does not report results on PoseTrack21, we rerun their method and reproduce their original predictions (which match the numbers reported in their paper on their PoseTrack18 evaluation). While it performs well on the incomplete PoseTrack18 setting, we observe many missed detections which result in a lower MOTA on PoseTrack21 where the full annotations are available. We dig further into this in Table 2, observing that we can match the numbers of 4D Humans in the original PoseTrack18 setting with CoMotion by applying a stricter threshold on detection confidences to dismiss a larger number of (correct) detections.

When evaluated on PoseTrack21 we observe that our 'strict' model is better than 4D Humans across the board. When we replace the strict threshold with our default, the system is heavily penalized in PoseTrack18, since many correct tracks are regarded as false positives, lowering MOTA from 59.9 to 51.3. But when these same predictions are evaluated on the more complete annotations provided by PoseTrack21, MOTA *increases* from 61.8 to 71.4.

**Pose estimation.** 3D pose estimation is typically assessed in an 'oracle' single-person setting where the bounding box of the target individual is known in advance. For a fair comparison to prior work using standard pose estimation metrics, we run CoMotion on individual frames and tight bounding-box crops while also reporting performance in the much more difficult setting where the full image is provided as input without an oracle conditioning signal.

We evaluate both 2D and 3D metrics, since 2D pose datasets offer more diverse in-the-wild data with challenging poses. Specifically, we report Percentage of Correct Keypoints (PCK) on COCO (Lin et al., 2014) and PoseTrack18 (Andriluka et al., 2018), and Mean Per-Joint Position Error (MPJPE) on 3DPW (von Marcard et al., 2018). PCK calculates the percentage of estimates whose distance to the ground truth falls under a given threshold, while MPJPE is the mean distance between 3D points after centering around the pelvis. PA-MPJPE performs an additional Procrustes Alignment step first. The results are summarized in Table 3. We follow the same setup as prior work, with the same bounding boxes, cropping, and resizing of each subject. Note that no video information is used here. As opposed to other methods, our model outputs multiple detected poses given an input image; we return the detection whose IOU is highest with the provided bounding box.

On oracle-cropped inputs, we observe similar pose estimation performance to 4D Humans, particularly on PoseTrack, and much stronger performance on 3DPW. We attribute the improvement on 3DPW to the updated pseudo-labels from NLF (Sárándi & Pons-Moll, 2024). Unfortunately, the quality of the pseudo-labels from NLF is worse on close-ups of people, which affects the performance of our model on such images. This is particularly pronounced on COCO, which contains a much higher percentage of close-ups than PoseTrack (which generally consists of images featuring full figures).

To assess the quality of CoMotion's pose estimates in a more practical setting, we further evaluate it on full images (bottom rows of Tab. 3). Since this setting no longer provides an explicit signal about a person's location and size within the image, we would expect a considerable drop in performance.

Surprisingly, the effect on pose accuracy is modest. We find that despite tackling the much more complex task of detecting multiple people and estimating their poses simultaneously, CoMotion estimates poses at a similar level of accuracy to a state-of-the-art top-down single-person method on oracle-crops.

Table 3: **Pose estimation.** Normalized PCK accuracy on projected 2D keypoints at varying thresholds on the COCO and PoseTrack datasets, alongside MPJPE of 3D keypoints on the 3DPW dataset. We highlight that our model performs similarly when provided the full image as input rather than an oracle-resized crop around a target person. See text for analysis.

| | Method | COCO | | PoseTrack | | 3DPW | |
|---|---|---|---|---|---|---|---|
| | | PCKn@0.05↑ | PCKn@0.1↑ | PCKn@0.05↑ | PCKn@0.1↑ | MPJPE↓ | PA-MPJPE↓ |
| oracle crop | PyMAF (Zhang et al., 2021a) | 0.68 | 0.86 | 0.77 | 0.92 | 92.8 | 58.9 |
| | CLIFF (Li et al., 2022) | 0.64 | 0.88 | 0.75 | 0.92 | 69.0 | 43.0 |
| | PARE (Kocabas et al., 2021) | 0.72 | 0.91 | 0.79 | 0.93 | 82.0 | 50.9 |
| | PyMAF-X (Zhang et al., 2023a) | 0.79 | 0.93 | 0.85 | 0.95 | 78.0 | 47.1 |
| | HMR 2.0a (Goel et al., 2023) | 0.79 | 0.95 | 0.86 | 0.97 | 70.0 | 44.5 |
| | HMR 2.0b (Goel et al., 2023) | **0.86** | **0.96** | **0.90** | **0.98** | 81.3 | 54.3 |
| | CoMotion | 0.79 | 0.93 | **0.90** | 0.97 | 63.6 | **36.1** |
| full | CoMotion | 0.79 | 0.92 | 0.88 | 0.96 | **60.0** | 37.3 |
| | - detection only | 0.79 | 0.91 | 0.88 | 0.95 | 60.5 | 37.3 |

Figure 3: We compare predictions made by CoMotion and 4D Humans unrolled through time on a sample from PoseTrack. Due to making independent predictions per frame, we observe that 4D Humans occasionally makes abrupt changes to the estimated pose (see green track on the right).

By default, we consider the standard CoMotion output to be the result of running both the detection and pose update modules. For completeness, we also report accuracy when using the outputs of the detection module only (last row of Tab. 3). While the additional update step slightly improves poses, we find that the detection head is already quite accurate and can be used standalone for single image multi-person pose estimation.

**Analysis.** CoMotion substantially outperforms the state of the art on multi-person tracking, while also yielding state-of-the-art 3D pose estimates. At the same time, CoMotion is an order of magnitude faster than 4DHumans, the strongest comparable system in the literature (Goel et al., 2023). Beyond the numerical results, there are qualitative differences in the behavior of CoMotion and 4DHumans. On challenging or truncated poses, 4DHumans sometimes exhibits high-frequency, abrupt changes as it jumps between possible interpretations of a given track from frame to frame. In contrast, CoMotion is more temporally coherent due to its recurrent predictions across time (Figure 3).

## 5.1 CONTROLLED EXPERIMENTS

We conduct several controlled experiments on architecture and dataset decisions. The results indicate that dropping the GRU modestly hurts performance, while removing the hidden state altogether is worse. On PoseTrack, ID switches are about 15% higher with no hidden state and pose accuracy drops by a couple points.

We also find that the third stage of the training curriculum does not substantially affect overall pose performance but leads to a notable improvement in tracking metrics. An interesting detail here is that the baseline after stage two of training is already quite strong. In large part this is probably due to the fact that occlusions that require reasoning over longer temporal windows are rare in our validation setting. That said, the reduction in ID switches after finetuning on longer sequences does suggest some meaningful change in model behavior. For tables and a more thorough review of these results we refer the reader to Appendix A.2.

## 6 CONCLUSION

CoMotion performs joint multi-person 3D pose tracking from monocular video. It operates online, processing each incoming frame in a streaming fashion. By unrolling predictions across time and allowing the network to jointly attend to all pose tracks and the full image context, CoMotion maintains accurate and temporally stable tracking and robust inference through occlusion.

There is still a lot of room for CoMotion to improve. One remaining failure mode is tracks that collapse together into a single identity. There are also occasional surprising identity switches where the network abruptly shifts a track to a different person. It is clear that the model lacks a notion of physicality to ground tracks in realistic ways.

Many aspects of the system would benefit from scaling, including model size, training time, and input resolution. A significant impediment to progress is the lack of high-quality video training data. We observe some initial positive results from adding pseudo-labeled forms of PoseTrack and DanceTrack to this work, but this would benefit from further investigation. In particular, this first attempt at pseudo-labeling misses out on annotations when people are in close proximity, and this is arguably an area that would be most important to supervise well for this class of model.

Another issue is the train-test discrepancy experienced by CoMotion. The model is exposed to fixed short clips with a fixed number of people during training but is then run in a setting where it can be unrolled for hundreds of frames in which tracks are constantly being added and deleted. This would be nontrivial to implement as a proper training stack, but there would be substantial benefits to doing so. One opportunity would be to train a differentiable track management module that would replace the track management heuristics used in this paper.

Another promising direction is to also model camera motion. An emerging body of work models 3D poses in a consistent world coordinate frame (Shin et al., 2024; Wang et al., 2024; Kocabas et al., 2024; Yin et al., 2024). We believe that decoupling the camera from the motion of people in the scene can increase robustness to extreme camera motion. Furthermore, CoMotion does not currently reason about absolute scale, thus 3D poses are not grounded in a shared extrinsic world frame. This is most salient with children, who are not modeled well by the SMPL parameterization and are placed further into the distance as larger adult-sized humans.

CoMotion provides a clean high-performing starting point for exploring these and other ideas. We will release the implementation and the trained model upon publication.

**Acknowledgements.** We thank Benoit Landry and Yurong You for early contributions to the project.

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

# A    APPENDIX

## A.1    TRACKING EVALUATION DETAILS

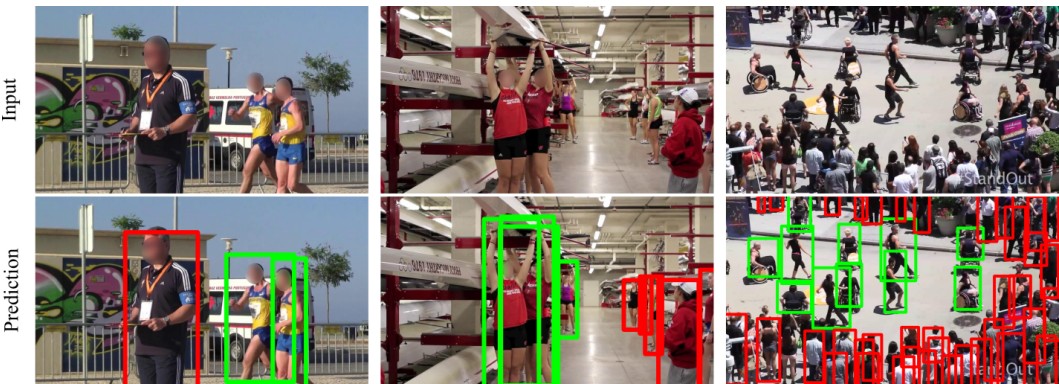

Figure 4: **Incorrect handling of missing annotations in PoseTrack18.** Due to incomplete annotations in PoseTrack18, predicted tracks may be incorrectly regarded as "false positives". We show representative samples where annotations are green and "false positives" are red.

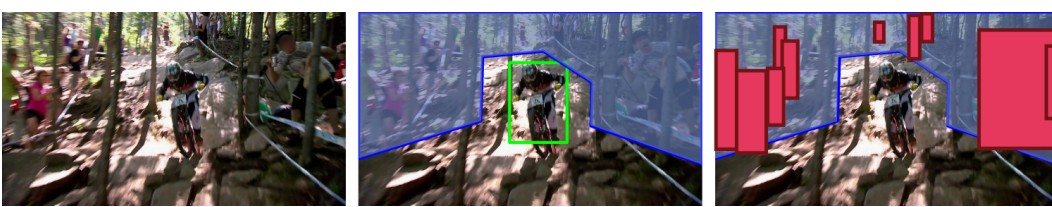

Figure 5: **Incorrect handling of missing annotations in PoseTrack 21.** PoseTrack21 addresses the incompleteness of PoseTrack18 annotations by providing 'ignore' regions to accompany the annotated tracks. For the frame on the left, the center image illustrates the annotation of the person in the center (shown in green) and a polygon defining the 'ignore' region in blue. The right image shows predicted tracks in red, which are still penalized as false positives by the PoseTrack21 evaluation code despite being contained in the 'ignore region'. This is a bug that we fix.

Tracking evaluation metrics, such as MOTA, are sensitive to false positives. A MOTA score lies in $(-\infty, 100]$, where 100 is a perfect score; we observe on the scene illustrated in Figure 5, if false positives are not ignored, our method garners a MOTA of around -1500.

Figure 4 visualizes the problematic handling of missing annotations in PoseTrack18. Due to missing annotations, correctly predicted tracks are labeled as false positives, which may produce misleading results.

To address this problem, Doering et al. (2022) introduced PoseTrack21 with more comprehensive annotations. PoseTrack21 provides keypoints for many more people and marks regions with missing annotations through "ignore regions". Unfortunately, the provided evaluation code contains a bug that still allows for incorrect handling of missing annotations. Figure 5 visualizes the behavior of the PoseTrack21 evaluation. Akin to PoseTrack18, predicted boxes in the ignore region are treated as false positives.

The issue has to do with the calculation determining whether or not to ignore a predicted box. The evaluation code looks at the IOU between the ignore region and the box and discards any box with an IOU greater than 0.1. Unfortunately, when the ignore region is particularly large, most boxes do not intersect a large percentage of the region's area. The IOUs in the example above vary between 0.01 and 0.07 – not enough to be discarded.

We make a one-line change to the PoseTrack21 evaluation code for our "fixed" evaluation: instead of IOU, we calculate the percentage of the bounding box that overlaps with the ignore region. This

Table 4: **Hidden state ablation.** From our full system (first row) we remove either the GRU (second row) or the hidden state altogether (third row).

| Hidden | GRU | PoseTrack | | | | EgoHumans | | 3DPW (val) | |
|--------|-----|-----------|---|---|---|-----------|---|------------|---|
| | | 2D@0.05↑ | MOTA↑ | IDF1↑ | IDs↓ | 3D@0.1↑ | MPJPE↓ | 3D@0.1↑ | MPJPE↓ |
| ✓ | ✓ | **74.9** | **70.9** | **78.6** | **491** | **77.4** | **83.8** | 90.4 | 61.3 |
| ✓ | - | 73.0 | 69.8 | 76.9 | 506 | 76.3 | **83.8** | 90.5 | 62.0 |
| - | - | 72.6 | 70.1 | 77.2 | 559 | 72.7 | 89.2 | 90.4 | **61.2** |

results in the expected evaluation behavior on scenes with ignore regions. We adjust the discard threshold from 0.1 to 0.3, so this means if 30% of a bounding box overlaps into an ignore region it will be discarded.

There are a few minor quirks that still remain in the annotations and evaluation. In some clips, ignore regions appear and disappear, and occasionally completely overlap with ground-truth annotations. We observe in the evaluation code by the PoseTrack21 authors that there is a mechanism to recover predictions that are matched to ground-truth tracks that have been filtered out by the ignore region. Perhaps this is to address the samples that have been both marked as ground-truth boxes and covered in ignore regions. Overall, these present a modest amount of corner cases and do not adversely affect the evaluation in the same way as the ignore region IOU calculation.

We would like to acknowledge that benchmarking and annotation work are seriously undervalued by the community and we appreciate the incredible work that went into producing both versions of PoseTrack. The above notes are just to provide clarity on what goes into calculating these metrics and the subtle ways in which unexpected and undesired behavior can slip through.

## A.2 CONTROLLED EXPERIMENTS

We perform additional experiments to investigate the impact of major design decisions on the final performance of the proposed approach. We report results on the PoseTrack validation set, on the 3DPW validation set, and on a subset of clips from EgoHumans (Khirodkar et al., 2023).

With EgoHumans, we do not follow an official evaluation setting but create our own evaluation by sampling some of the exocentric camera clips from the Lego, Tagging, and Fencing scenes. We find EgoHumans interesting as one of the few datasets with 3D annotations and complete annotations for multiple people. Critically, there are annotations through occlusion because data was collected in a multi-view setting. Most other evaluation datasets do not have ground truth for occluded and truncated samples.

All evaluation is done across video in the multi-person setting with full images resized and padded to 512x512. For the purpose of this evaluation we do not manage tracks. On the first frame, we choose detections that match closest to the corresponding ground truth annotations and then unroll the update step without any further modifications to the set of tracks. The evaluation assumes that each unrolled track will match the annotations corresponding to the person assigned from the first frame. On PoseTrack, we unroll 16 frames, and on 3DPW and EgoHumans we unroll 48 frames.

One note, for these ablations, the 2D PCK @ 0.05 on PoseTrack is calculated differently from the results using the HMR evaluation code reported in Tables 3. The normalization factor is more strict and thus not directly comparable. We use this metric only in the context of ablations comparing different versions of our model.

**Hidden state.** We assess the importance of the recurrent hidden features through an ablation study. We report detailed metrics in Table 4. The first row represents our full approach, the second row drops the GRU but preserves a learned hidden state that is passed through time, and the third row drops the hidden state altogether. Performance is slightly worse without the GRU, and even worse without any hidden state at all.

Without a hidden state, the only information passed across consecutive frames are the previous SMPL poses. For the vast majority of tracking situations this is sufficient to know which pose corresponds to which person, but can lead to failures in the face of occlusions and similar poses in similar positions.

Table 5: Ablation study on the impact of PoseTrack and DanceTrack.

| PoseTrack | DanceTrack | PoseTrack | | | | EgoHumans | | 3DPW (val) | |
|---|---|---|---|---|---|---|---|---|---|
| | | 2D@0.05↑ | MOTA↑ | IDF1↑ | IDs↓ | 3D@0.1↑ | MPJPE↓ | 3D@0.1↑ | MPJPE↓ |
| ✓ | - | 73.2 | 68.4 | 75.6 | 576 | 76.5 | 84.2 | 90.0 | 62.8 |
| - | ✓ | 72.4 | 69.8 | 75.6 | 703 | **77.5** | 84.7 | **90.6** | 61.9 |
| ✓ | ✓ | **74.9** | **70.9** | **78.6** | **491** | 77.4 | **83.8** | 90.4 | 61.3 |

Table 6: Impact of additional training stage on longer sequences.

| Stage 2 | Stage 3 | PoseTrack | | | | EgoHumans | | 3DPW (val) | |
|---|---|---|---|---|---|---|---|---|---|
| | | 2D@0.05↑ | MOTA↑ | IDF1↑ | IDs↓ | 3D@0.1↑ | MPJPE↓ | 3D@0.1↑ | MPJPE↓ |
| ✓ | - | **74.9** | 70.9 | 78.6 | 491 | 77.4 | 83.8 | **90.4** | **61.3** |
| ✓ | ✓ | 74.6 | **71.4** | **79.5** | **382** | **77.9** | **83.2** | 90.1 | 61.9 |

We observe that without a hidden state, overall pose accuracy on videos from EgoHumans and PoseTrack drops, accompanied by a notable decrease in tracking performance. Removing the hidden features increases ID switches by almost 15%. In contrast, 3DPW performance is largely unchanged which makes sense as the videos in the validation set consist mostly of one person, sometimes two, so tricky tracking situations do not arise in this context.

**Pseudolabeled video datasets.** We report results in Table 5 after removing either PoseTrack or DanceTrack our two real world datasets that were pseudolabeled for video training to see the role, if any, they have on performance. These results are calculated after the stage 2 training on 8 frame sequences, not the additional longer sequence finetuning. We observe a modest effect on simpler scenes like 3DPW and EgoHumans that only consist of up to four people with static cameras. A more measurable drop in performance occurs on PoseTrack if either dataset is excluded, particularly in the tracking results (note the number of ID switches). PoseTrack itself seems to have a larger impact which would make sense as it is in domain for its own validation set while DanceTrack consists of a fairly different video distribution.

**Curriculum.** We examine how much performance improves with an additional training stage on longer sequences. After stage 2 the model has only ever been trained on clips up to 8 frames in length. In some ways, it is impressive that it does as well as it does having never been exposed to sequences longer than a quarter of a second. The pose accuracy does not change much with additional finetuning, but there are notable gains in tracking performance (Table 6).

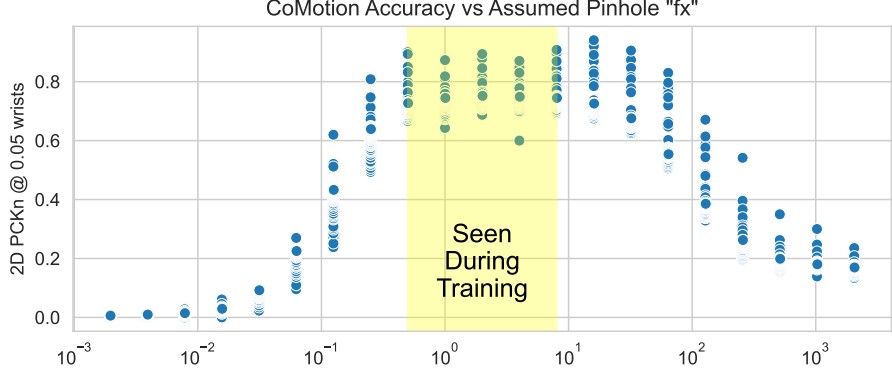

Figure 6: **Impact of the assumed focal length.** To investigate how the focal length of the intrinsics matrix affects performance we run our model on PoseTrack videos (for which we do not have ground truth camera calibration) and report 2D PCK accuracy. We adjust the assumed focal length and observe that the network's 2D keypoint accuracy is consistent as long as we remain within a realm of values which correspond to what one would typically find with most camera hardware apart from extremely wide-angle options such as a fish-eye lens. In the above figure, fx (the x-axis) is normalized by the image width.

**Sensitivity to provided camera intrinsics.** To assess the sensitivity of our model to the input camera focal length, we evaluated CoMotion on PoseTrack videos (for which we do not have ground truth camera calibration) at various focal lengths. Figure 6 visualizes the results. We find that the network's 2D keypoint accuracy remains consistently high across a wide range of inputs.

### A.2.1 RUNTIME

We measure the runtimes of CoMotion and baselines on the PoseTrack21 validation set and report results in Figure 7. All measurements were made on the same hardware using the code provided by the authors of each method. We measure the time to run the complete tracking stack unrolled across all PoseTrack validation videos. We find that CoMotion significantly outperforms prior work. Specifically, CoMotion is approximately 1.4x faster than PARE (176ms vs 258ms) and 12x faster than 4D Humans (176ms vs 2163ms) on average.

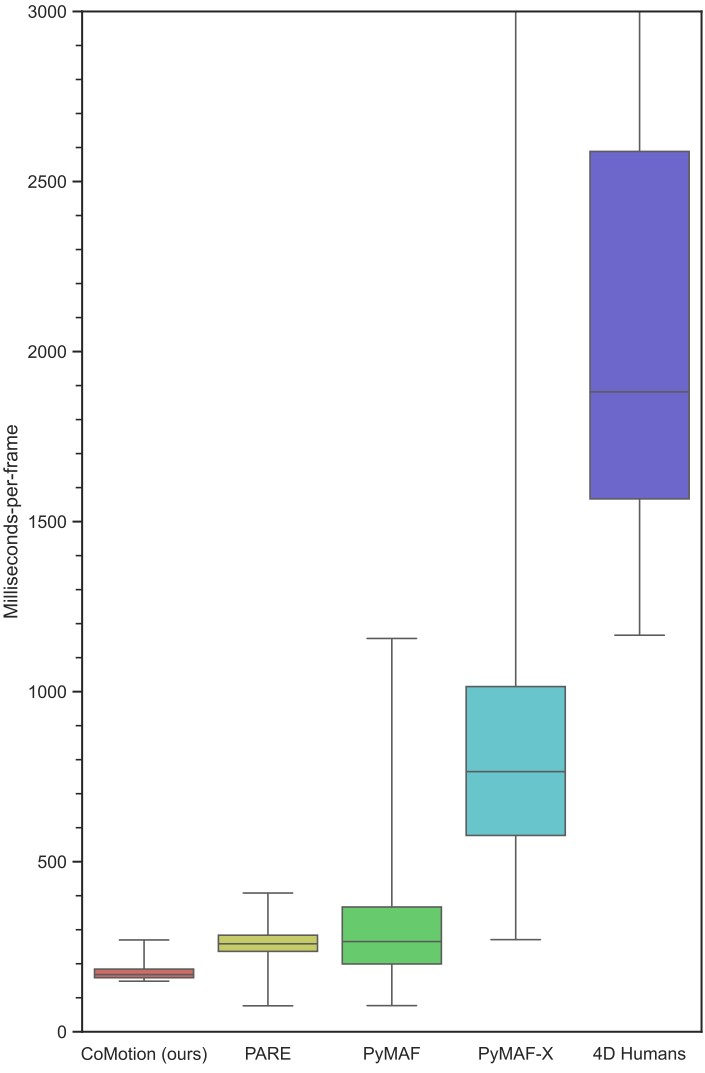

Figure 7: Comparing the per-frame runtime of CoMotion with prior work on the PoseTrack21 validation set. All measurements were made on a V100 GPU using the code released by the respective authors. CoMotion is significantly faster than prior work. Specifically, CoMotion is approximately 1.4x faster than PARE (176ms vs 258ms) and 12x faster than 4D Humans (176ms vs 2163ms) on average.

```python
def get_image_features(image):
    # Run ConvNextV2 stages
    x = convnext.stem(image)
    feat_pyramid = []
    for stage in convnext.stages:
        x = stage(x)
        # Linearly project image features from each stage
        feat_pyramid.append(conv(x))

    # Upsample low-res features (using einops notation)
    upsample_pattern = "... (c h2 w2) h w -> ... c (h h2) (w w2)"
    feat_pyramid[2].rearrange(upsample_pattern, h2=2, w2=2)
    feat_pyramid[3].rearrange(upsample_pattern, h2=4, w2=4)

    # Fuse into single tensor at 1/8th input resolution
    image_features = sum(feat_pyramid)
    image_features = layer_norm(image_features)

    return image_features
```

Figure 8: Pseudocode for the image encoder.

## A.3 ADDITIONAL METHOD DETAILS

### A.3.1 IMAGE ENCODER

We use the ConvNextV2 (Woo et al., 2023) implementation provided in the timm library (Wightman, 2019). As with most convolutional backbones, features are extracted in consecutive stages to progressively lower resolutions. There are four stages yielding features at 1/4th, 1/8th, 1/16th, and 1/32nd of the input image resolution. As a simple strategy to fuse early and late features, we linearly project the outputs from each stage to a feature tensor at 1/8th the input resolution. For example, a 2x2 convolution with stride 2 is used on the highest resolution features, while a 1x1 convolution is applied to the lower resolution feature maps with an output of either 4 or 16 times the number of feature channels which can be reshaped into an upsampled tensor. The resulting features are added together to produce a feature tensor $F^t$ for subsequent stages. We refer to Figure 8 for PyTorch-like pseudocode outlining these steps.

We find performance improves if we add some signal about intrinsics to the features $F^t$. We apply rotary positional embeddings to a fraction of the channels to encode the $u$ and $v$ values in pixel space and the corresponding $x$ and $y$ values in world coordinate space. This yields a modest bump in keypoint accuracy.

### A.3.2 DETECTION STEP

**Architecture.** We perform detection using a single-shot detection approach (Liu et al., 2016). We depict the architecture in Fig. 9. Given the output features $F^t$ from the image encoder, we define a detection head that applies several consecutive downsampling and ConvNeXt blocks (Woo et al., 2023) to yield a low-resolution feature pyramid. At each feature scale we apply 1x1 convolutions to decode candidate detections. The model produces multiple candidate poses at each spatial location. The number of poses per location is a hyperparameter, and we found 4 to provide a good balance of performance and compute overhead.

The model outputs over a thousand candidate detections for an input 512x512 image before non-maximum suppression. Specifically, the feature pyramid is pooled down 32x, 64x, and 128x from the input image resolution, so for a 512x512 input image, output detections are produced on 16x16, 8x8, and 4x4 grids. With 4 output poses per position in each grid, the total pool of candidate detections is 1,344 for a given image.

**Output representation.** For each detection, the network produces SMPL parameters (translation, root orientation, body pose, betas) as well as a confidence term. For the body pose, rather than produce joint angles directly, the network predicts a latent embedding that is passed to a separate

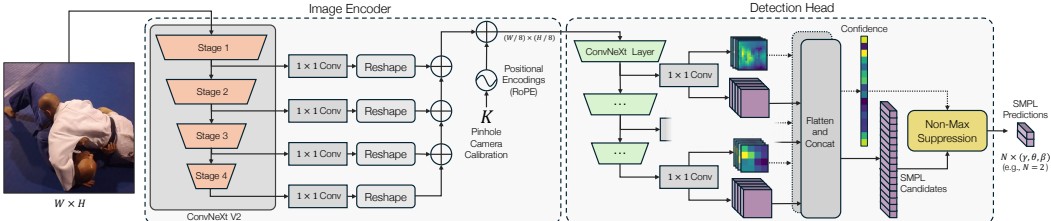

Figure 9: **Architecture of the image encoder and detection step.** We visualize the core modules involved in producing encoded image features and per-frame detections. The *image encoder* produces a multi-resolution feature pyramid using the intermediate activations from ConvNeXtV2, ranging from $^1/_8$ to $^1/_{64}$ the input resolution. The features' channel dimensions are then expanded with 1x1 convolutions such that they can all be reshaped to $\frac{1}{8}$ resolution and summed. The image encoder also applies positional encodings derived from the uncalibrated pixel coordinates. Next, the *detection head* produces a large set of candidate SMPL parameters and associated confidences, derived from feature maps at three spatial resolutions ($\frac{1}{8}, \frac{1}{16}, \frac{1}{32}$). We apply non-maximum suppression to return the final set of detections.

MLP decoder that maps to the output axis-angle pose representation. This decoder is shared in the update step as well.

We take measures to ensure the model's predictions project to the image correctly given the provided input intrinsics matrix $K$. An output on the grid at row $i$ and column $j$ is placed at a corresponding base pixel position $(u_j, v_i)$, the model predicts x and y offsets in pixel space $(du, dv)$. The depth is predicted in log-space $(z_{log})$ and is scaled by the focal length $f_x$ and a hyperparameter $z_{default}$. We lower the default depth at each level of the pyramid so that at coarser spatial resolutions, candidates are initialized closer to the camera. The SMPL translation term $\gamma$ is then computed as follows:

$$u = u_j + du \tag{1}$$

$$v = v_i + dv \tag{2}$$

$$z = z_{default} * f_x * \exp(z_{log}) \tag{3}$$

$$\gamma = z \cdot \begin{bmatrix} f_x & 0 & c_x \\ 0 & f_y & c_y \\ 0 & 0 & 1 \end{bmatrix}^{-1} \begin{bmatrix} u \\ v \\ 1 \end{bmatrix} \tag{4}$$

In summary, we have the following network outputs:

- $\gamma$ *(translation)*: $z_{log}$ and pixel x, y offsets $(du, dv)$ applied relative to the output pixel position. Together these terms are used to derive $\gamma$ which is the translation in camera coordinate frame.

- $\theta$ *(pose)*: The SMPL pose angles are the concatenation of the 3DOF root orientation, and body pose joint angles which are decoded from an MLP applied to the output 256 dimensional pose embedding from the detection head.

- $\beta$ *(shape)*: 10-dimensional SMPL betas $\beta$ that determine body shape and proportions.

- *confidence*: Output confidence of the detection.

If a particular detection is chosen to serve as a new track instance we define a track state $X = (\gamma, \theta, \beta, h)$ from these outputs. The value $h$ refers to an additional latent hidden state for the track, this is not predicted by the detection head and instead is initialized with a vector of all zeros.

**Supervision.** It is is nontrivial to define a predetermined strategy to assign ground truth annotations to a particular candidate output at a specific spatial location. Instead, we flatten and concatenate all candidate detections and assign them to the ground truth detections with Hungarian matching. We compute the OKS between the 2D projected keypoints of detections and ground truth annotations and adjust the score by the confidence of the detection so that confident detections are more likely to be matched and have a loss applied to them.

```python
def cross_attention(image_key, image_value, tokens):
    # Image key and value shape
    # B: batch_size, H * W: height x width, C: feature_dim

    # Input tokens shape
    # B: batch_size, N: num_people, M: num_tokens, C: feature_dim

    # Perform cross attention on union of all tokens for all people
    q = token_to_query(tokens)
    q.rearrange("b n m c -> b (n m) c")
    px_feedback = F.scaled_dot_product_attention(
        q, image_key, image_value
    )
    px_feedback.rearrange("b (n m) c -> b n m c")
    tokens = tokens + post_attention_mlp(px_feedback)

    # Additional transformer with attention across people
    tokens.rearrange("b n m c -> (b m) n c")
    tokens = cross_people_attention(tokens)

    # Additional transformer with attention per-person
    tokens.rearrange("(b m) n c -> (b n) m c")
    tokens = per_person_attention(tokens)
    tokens.rearrange("(b n) m c -> b n m c")

    return tokens

def update(K, image_features, smpl_params, hidden):
    # Encode tokens
    pred_2d = get_2d_projection(K, smpl_params)
    tokens = encode_2d_to_tokens(pred_2d)
    tokens += encode_smpl_to_tokens(smpl_params)
    tokens += encode_hidden_to_tokens(hidden)

    # Prepare image keys and values
    image_tokens = cat([image_features, positional_encoding], 1))
    image_tokens.rearrange("b c h w -> b (h w) c")
    image_key = linear(image_tokens)
    image_value = linear(image_tokens)

    # Apply cross-attention
    tokens = cross_attention(image_key, image_value, tokens)
    tokens = cross_attention(image_key, image_value, tokens)

    # Decode outputs
    hidden = gru_update(tokens, hidden)
    smpl_params = decode_update(tokens, smpl_params, hidden)
    return smpl_params, hidden
```

Figure 10: Pseudocode for the update step.

For the matched detections we apply losses to the 2D reprojection of their 2D keypoints as well as SMPL losses on the joint angles, root-centered 3D keypoints, and in the case of BEDLAM to the ground-truth betas. We also apply a binary cross entropy loss to the confidence term, using the matched samples as positives and a random subset of the unmatched samples as negatives. At test time, we do standard non-maximum suppression using OKS instead of bounding box IOU to get the final set of detections.

### A.3.3 UPDATE STEP

The update step attends to the latest input image to provide new poses for all tracked people in the scene. To perform the update, first we encode the previous state $\mathbf{X}^{t-1}$ into a set of tokens $x \in \mathbb{R}^{N \times M \times D}$ where $N$ is the number of tracked people, $M$ is the number of tokens used to represent an individual track, and $D$ is the feature dimension. We can set $M$ to any value to control the memory and compute overhead of the update step, and for all experiments we set $M = 24, D = 512$. To get $x$, we take the current SMPL parameters and compute 2D projected keypoint locations, then pass the raw SMPL parameters, the 2D keypoints, and the current hidden state for that person through an MLP to produce $M$ tokens, batching this operation and concatenating to produce the full $N \times M$ set.

We then attend to image tokens which are produced by flattening the image features $F^t$ and concatenating them with sinusoidal positional embeddings. We apply linear layers to get per-pixel key and value embeddings which are attended to with queries from the per-person tokens $x$. The output of the cross-attention operation gets added residually to $x$, and we apply additional transformer layers for further processing. We then pool across the $M$ tokens for each person and apply an MLP to predict the new output poses and hidden states. This entire process is outlined in pseudo-code provided in Figure 10.

The goal of the above steps is to have a flexible way to operate on an unordered discrete set of tracks and efficiently attend to dense per-pixel image information without explicitly defining what information might be relevant for which tracks. We use standard attention as the building block upon which we exchange information between image features and the tracks themselves yielding a clean and straightforward update to all poses.

**Output representation.** The output follows the same output pattern of the detection head. The breakdown of outputs is as follows:

- $\gamma$ *(translation):* We output $(z_{log}, du, dv)$ and follow the same calculation for $\gamma$ as in the detection stage (Sec. A.3.2). In this case, the pixel offsets $(du, dv)$ are added to the projected location of $\gamma_{t-1}$.

- $\theta$ *(pose):* The network predicts a new root orientation and pose embedding that is passed through the same MLP decoder as in the detection step. If we instead ask the network to learn residual updates to either the root orientation or pose embedding, the network struggles when transitioning between rapid rotations or rare and seldom seen poses.

- $\beta$ *(shape):* We do not update the SMPL betas at all. When allowed to do so, the network would sometimes opt to make a person larger or smaller rather than physically moving them towards or away from the camera. Results were more sensible and stable with a fixed body shape across time.

### A.3.4 MODIFIED OKS

At various stages of our pipeline, we need to measure the similarity between two poses (e.g., matching detections to ground truth annotations, matching detections to existing tracks). We perform these comparisons on their projected 2D keypoint locations. We choose 2D keypoints rather than SMPL pose parameters or other 3D cues for compatibility with datasets where only 2D keypoints are available. We follow the COCO OKS ("Object Keypoint Similarity") calculation found at cocodataset.org/dataset/#keypoints-eval which returns a normalized value from 0 to 1 even with a variable number of valid keypoints:

$$\text{OKS} = \frac{\sum_i v_i \cdot \exp(-d_i^2/2s^2\kappa_i^2)}{\sum_i v_i} \tag{5}$$

In the above equation $d_i$ is the Euclidean distance between the pair of keypoints corresponding to the $i$th body part, while $v_i$ is a binary value indicating whether that part is valid. The standard deviation of the Gaussian is defined by the object scale $s$ and $\kappa_i$, a predefined constant to adjust the deviation per-keypoint. We make some slight modifications to the above calculation to better suit our needs:

$$\text{OKS}_{ours} = \frac{\sum_i v_i \cdot (1 + d_i^2/2s^2\kappa^2)^{-1}}{\sum_i v_i} \tag{6}$$

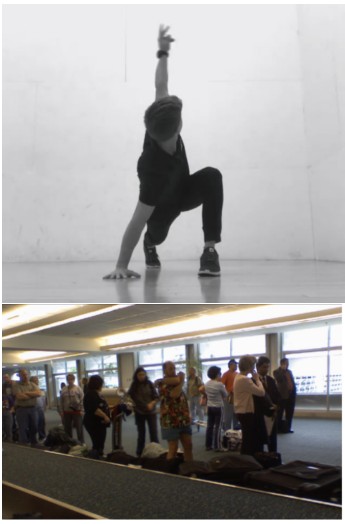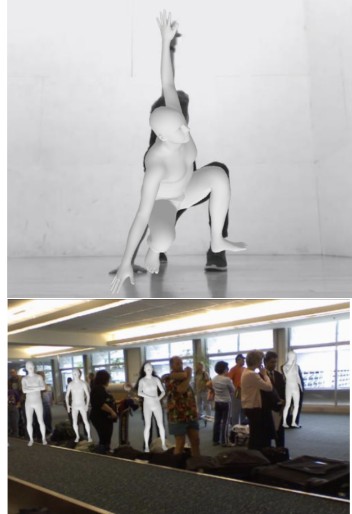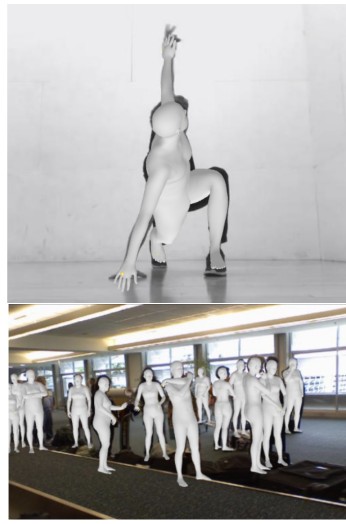

Figure 11: **Comparison of pseudolabeled annotations.** Left: input image. Middle: original pseudolabels provided by Goel et al. (2023). Right: the new pseudolabels produced by running NLF (Sárándi & Pons-Moll, 2024). We observe better head and foot correspondence with the new outputs *(top)* and more comprehensive annotations across people in the scene *(bottom)*.

We use a single constant $\kappa$ instead of per-keypoint values, and replace the Gaussian with a Cauchy distribution which has a heavier tail. This is important as we often compare points that are far from each other and need a nonzero signal at those larger distances. Finally, another important detail is that we do not have a ground-truth scale $s$ as would be used in the COCO evaluation, so we calculate a scale term on the fly based on the projected 2D bone lengths of each set of keypoints:

$$s = \max_{(i,j) \in L} \left( \frac{\|p_i - p_j\|}{l_{ij}} \cdot v_i \cdot v_j \right) \tag{7}$$

Limbs are defined by a reference of pairs of adjacent keypoints $L$ and default limb lengths $l$, we calculate the distance between the two endpoints of the limb $p_i$ and $p_j$ and compare this to the reference distance $l_{ij}$ yielding a ratio of the size of the current projected limb compared to our baseline size. We take the max over all ratios to account for any limbs that are foreshortened. The largest value typically correlates well with the perceived size of the person in the image. While only a rough approximation, this signal provides a more reliable indicator of the size of a person as opposed to alternatives based on bounding box size. When comparing two sets of keypoints we will use the larger of the two scales, or in the case where we are comparing to a ground-truth annotation we use the scale of the annotation.

### A.4 PSEUDOLABELED DATA

Training on large-scale in-the-wild pseudolabeled data is crucial for achieving accurate 3D pose estimates on the most challenging poses (Goel et al., 2023). But pseudolabeling is imperfect. For example, many of the samples in the data made available by the authors of 4D Humans has odd behavior when people are truncated which includes raised hands and feet at the image boundaries. Models trained on this data inherit these quirks. More importantly, we use this data for training our model to perform detection, but the pseudolabeled data is quite sparse. So our model would learn to selectively ignore people in the scene which proved difficult to overcome with adjustments to losses and other training decisions.

A recent strong 3D pose model was released referred to as NLF (Neural Localizer Fields) (Sárándi & Pons-Moll, 2024) which offered an opportunity to produce new pseudolabels. We ran this model off-the-shelf on the full set of InstaVariety, COCO, and MPII images. With a strong detector, the images are much more comprehensively annotated than before and the quality of poses improved. In particular, there is better alignment of the head and feet to the image (see Figure 11).

We also investigated the use of pseudolabels on two video datasets: PoseTrack and DanceTrack. We leverage the existing track annotations provided by these datasets, and run the model conditioned on the provided dataset bounding boxes. This offers some of the only in-the-wild challenging multi-person 3D video data. Unfortunately there is one issue which is difficult to overcome, which is that bounding boxes are poor conditioning signals when two people are in close proximity. The output of the model by Sárándi & Pons-Moll (2024) will occasionally be heavily distorted or more typically, will be collapsed to a single person. This is one of the areas where it is most important to get the supervision right when training CoMotion. PoseTrack comes with 2D annotations that allowed us to catch some of these situations automatically, and we further develop a heuristic for flagging these cases and setting the annotations as invalid on DanceTrack.

An additional failure mode of the NLF model is that we occasionally run into odd outputs with truncated people. Fortunately, the model output includes reasonably well calibrated uncertainty terms which we use to automatically filter out bad samples. Even so, our system does inherit a bit of this behavior, so we find that after training, the results of CoMotion on close ups and truncated figures does not align quite as well as figures that are fully in frame.

Another motivation for pseudolabeling all of the real image and video datasets with this model was to provide a consistent output annotation format. We experimented with training on mixed annotation formats, but often ran into issues as the keypoints used to annotate 2D datasets like COCO and PoseTrack do not align perfectly with the corresponding SMPL keypoints. The differences can be subtle, but the mismatch in positioning of the hips and shoulders can lead to the network fitting poses in odd ways in order to minimize the 2D reprojection loss used during training.

This does have an effect when it comes to benchmarking across these datasets. The model trained on this new data achieves a significant improvement in 3DPW numbers (Table 7). Looking qualitatively at the outputs of models trained on these respective datasets (Figure 12), the result after training on the original pseudolabels are not bad, but the NLF-based pseudolabels align much better to the expectations of the 3DPW annotations.

Table 7: **Pseudolabeled data ablation.** Comparing 3DPW train performance between a model trained with the original pseudolabels from Goel et al. (2023) and on the new pseudolabels from Sárándi & Pons-Moll (2024).

|  | 3D@0.1↑ | MPJPE↓ |
|---|---|---|
| Original pseudolabels | 76.8 | 73.8 |
| New pseudolabels | **90.4** | **54.9** |

To summarize, we identified several key areas that motivated the switch to new pseudolabels:

- The original pseudolabels are sparse, often only accounting for one or two people in a crowded image which adversely affected our detection performance.
- The new pseudolabels better match people's heads and feet.
- The new pseudolabels improve performance on rare and very difficult poses.
- The new pseudolabels allow us to train on both image and video datasets with a uniform output format and not worry about mismatches in annotation styles.

As a trade-off we unfortunately regressed on closer shots of people, but this is something that could be addressed easily in the future with better data and is not a fundamental limitation of our method.

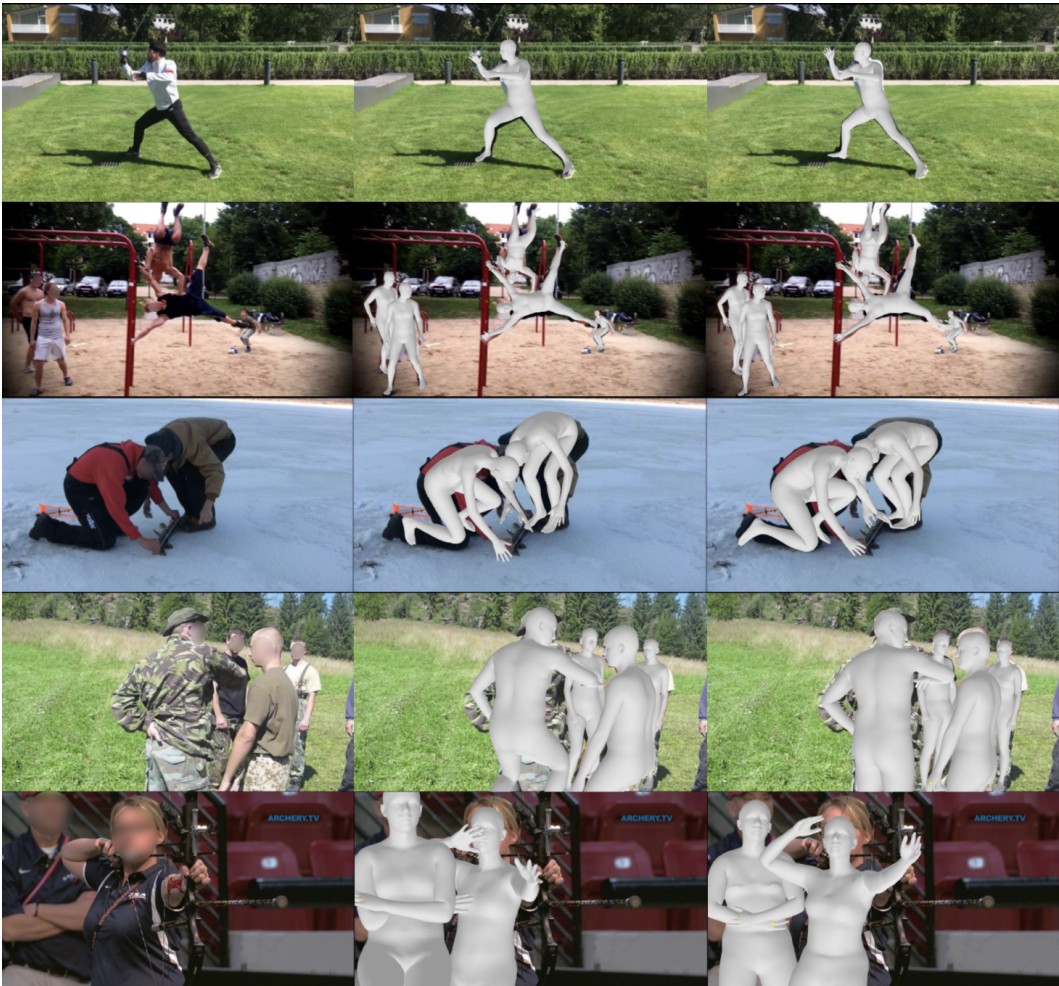

Figure 12: **Comparison of model predictions when trained with differing pseudolabels.** Left: input image. Middle: detections trained on original pseudolabels provided by Goel et al. (2023). Right: detections trained on new pseudolabels produced by running NLF (Sárándi & Pons-Moll, 2024). The effects are subtle, but the body fitting is improved and when tested in-the-wild we notice better accuracy on extreme poses like those seen in breakdancing and gymnastics clips. The one notable regression is on closeups, as illustrated in the bottom row.

