# OpenReview forum: "CoMotion: Concurrent Multi-person 3D Motion"
_ICLR.cc/2025/Conference — ICLR 2025 Poster_

### Official Review · Reviewer_5TGo · 2024-11-03

**Soundness:** 3
**Presentation:** 3
**Contribution:** 3
**Rating:** 6
**Confidence:** 5

**Summary:**

This paper describes a method to simultaneously track and regress coherent 3D human motion from videos with occlusion scenes. The key innovation is to update the 3D pose of multiple tracking targets from sequential images in a frame-by-frame manner. The baseline method for  comparison is 4DHuman, which tracks people via artificially associating the 3D human pose, location & appearance based on pre-defined matching rules. While the proposed method follows a tracking-by-attention way, which detects objects and updates existing tracklets via cross-attention between image and tracklet tokens. Query tokens of existing tracklets are firstly updated with current image tokens. The updated 3D human poses are matched with 2D bouding box detection of current frame. The unmatched detection will be initialized with new tracks and get updated to generate new 3D pose. Evalutation results on PoseTrack21 testified the effectiveness of the prospoed method. Compared with 4DHumans, the MOTA, IDF1, and IDR get significantly improved.

**Strengths:**

1. Investigating the tracking-by-attension way of mutl-person 3D human motion tracking in crowded scenes is very challenging and interesting. The results of this paper is quite encouraging and exciting.
2. Updating the tracklet state with assistance of bouding box detection is reasonable. Design of 3D human pose update module is great and seems to be very effective. Subtle 3-stage training process is brilliant.
3. The experiment details are sincerely and clearly explained.

**Weaknesses:**

1. The tracking experiment is mostly performed on PoseTrack dataset only. Maybe some challenging occlusion scenes in 3D benchmarks, such as MuPoTS-3D and 3DPW could testify the performance of the proposed method in a more comprehensive way.

2. The way of showing qualitative videos is inconvenient for reader to watch. One video with more attractive demos / comparisons would help a lot.

**Questions:**

See weakness part.

---

> ### Author Response · Authors · 2024-11-28
>
> **Qualitative videos**: We appreciate the feedback about the qualitative videos. We had hoped the accompanying HTML page would help compare results side-by-side, but before any release we will take that into account and prepare a clearer demonstration of our method’s results!
>
> **Benchmarking on 3DPW and MuPoTS-3D**: Thanks for the suggestion. We focused our efforts on PoseTrack since although it does not provide ground truth 3D annotations we can still evaluate 2D pose estimation quality and PoseTrack offers extremely challenging dancing and sports scenes with large crowds that test the limits of our pose estimation and tracking model. In contrast, MuPoTS, for example, generally consists of scenes of only 2-3 people typically walking around with a fixed camera. In our controlled experiments we made sure to also include quantitative evaluations on EgoHumans, which offers similar benefits to 3DPW/MuPoTS with a fair number of serious occlusions in its scenes.

---

> > ### Author Response · Authors · 2024-11-30
> >
> > Dear reviewer 5TGo. Please let us know if we have addressed your concerns. If you believe we have, could you please consider raising your score to reflect that?

---

### Official Review · Reviewer_VXUb · 2024-11-03

**Soundness:** 3
**Presentation:** 2
**Contribution:** 3
**Rating:** 6
**Confidence:** 2

**Summary:**

CoMotion introduces a complex system for multiple person tracking from a monocular camera, based on the tracking-by-attention paradigm. To overcome the challenge of limited data for pose estimation the authors train the model in multiple stages and add various datasets to gradually teach the model to track 3D persons across time.

**Strengths:**

The method solve a highly relevant problem, tracking multiple people from a monocular camera. The method performs well on tracking and pose estimation benchmarks and the authors identified and fixed a bug in existing benchmarks, which is very helpful to the research community.

**Weaknesses:**

The method makes use of various datasets and human pose modalities, such as SMPL, pseudo-SMPL and COCO. Could the authors describe how they handle different pose skeleton modalities, i.e. how do they train their SMPL pose estimation model with COCO skeleton supervision?

LL292-299: it is interesting that the authors do not add tracks during training: have the authors conducted an analysis of the training data to determine how many tracks appear / disappear on average? I would assume that this could potentially be very detrimental to the learning, as the model might learn to “ignore” certain people (i.e. if they were occluded before, or if they are at the edge of the image). Have the authors done any analysis of those kind of edge cases?

What do the authors mean by “[we] do not expect the shape of the mesh to correspond to the person’s physical body” (LL207-208)? The shape does not just model the mesh surface but also influences the bone lengths, so the skeleton is affected by it as well. Could the authors clarify this?

What do the authors mean by “the intrinsic matrix [does not] necessarily reflects the actual calibration of the camera“? Especially, as the authors claim in the next sentence that their method is robust to any provided intrinsic camera! Such a claim should be ablated, i.e. compare the system predictions with K provided and without K provided.

Can the authors elaborate on what they mean by “strict” in Table 2?

Figure 2 is not self-containing as it does not describe all elements of the image, i.e.e method part (A) is missing.

Typo in L205-206: the parameters $t$, $\beta$ and $\theta$ are missing from the SMPL parameter discretion.

LL324-325: the authors should provide more details on how they apply the regularization for the beta optimization.

— suggestions — (will not influence my rating)

The authors could utilize the CHI3D [a] as an additional benchmark as it comes with well-tracked 3D poses (SMPL-X) at higher quality than 3DPW and with more challenging human-human interactions.


[a] Fieraru, Mihai, et al. "Three-dimensional reconstruction of human interactions." CVPR 2020.

**Questions:**

L318: what is the “off-the-shelf MAE”?

I wonder if the authors have thought about utilizing a generative model as pose prior, ie. Following ScoreHMR. The system could then be understood as a Recursive Bayesian Filter, which would provide a nice mathematical foundation for the tracking algorithm.

---

> ### Author Response · Authors · 2024-11-28
>
> **Skeleton modalities**: How we handle skeleton modalities is a great question, and one that took up a considerable amount of time and attention when iterating on this project. We have added a section to the appendix providing more details. But to briefly summarize:
>
> The 2D keypoints in the COCO format are placed at different positions than the corresponding key points in the SMPL skeleton. This is noticeable in the hips where COCO-style annotations place the hips near the outer edges of the body higher along the waist line while SMPL puts the hips roughly at their anatomical joint position attached at the pelvis. Other subtle differences exist for ankle and shoulder joints. And a particularly challenging case is the handling of face keypoints, where it can be difficult to have the SMPL mesh fit the head such that it matches 2D-annotated eye, ear, and nose keypoints.
>
> We explored several approaches to handling the discrepancies. A common way of dealing with this discrepancy is to define linear regressors from SMPL mesh vertices out to joint locations. We attempted to use a predefined regressor from the SMPL mesh out to COCO-style keypoints as well as attempted to learn a custom regressor during training. We also investigated a simpler choice which was to modify the loss. Specifically we down-weighted the 2D reprojection losses of keypoints with a greater mismatch between annotation styles (e.g. face + hips). This was inspired by the benefits TokenHMR demonstrated with their threshold loss. In the end, we ended up not using any of these approaches, and instead manually narrowed the hips on the COCO-style annotations, and treated everything else the same as this worked best. One consequence is that on the pseudo-labeled datasets the 3D losses are applied on the pseudo-SMPL keypoints while the 2D reprojection loss is on the provided 2D keypoints which don’t necessarily match. Interestingly, 3DPW seems to have followed a similar strategy in their annotations where the SMPL hip keypoints match up with the detected 2D keypoints narrowed to half their original width.
>
> **“Ignoring” people**: In general we do not find that the model treats tracks instantiated later any differently from tracks instantiated at the beginning. As long as we provide a box to initialize a particular person, the model will predict the pose for that person and follow them. That said, one weak point in our tracking heuristics is instantiating a new track for someone who appears from behind an existing track. In this case, it is not until the person is sufficiently separated that they are likely to get instantiated.
>
> **Mesh shape**: Thank you, we revised this statement in the paper. We rely on supervision from pseudo-GT SMPL annotations which were not built with the intention of correctly matching body shape/size. An example for this is SMPL’s modeling of infants/children, which are typically predicted with the height and body proportions of an adult body. A good investigation into this is provided in the concurrent CameraHMR (https://arxiv.org/abs/2411.08128).
>
> We nevertheless predict the SMPL shape parameters as they determine bone lengths, which affects the model’s 3D pose accuracy. We do however not expect remaining shape properties like girth or relative proportions to be modeled correctly by the SMPL shape parameters and consequently do not expect to be competitive with any method trained for accurately estimating body shape.
>
> **Intrinsics**: Inference requires an input intrinsics matrix. If no ground truth is provided, we fall back to a reasonable default. As this default may not match the actual (missing) ground truth, we augment input matrices during training to increase robustness. We perform an analysis on PoseTrack PCK accuracy as a function of the assumed focal length and find that the network maintains good 2D accuracy across the full range of focal lengths seen during training. This can be found in Appendix A.2.5. We clarified in the revised paper.
>
> **Typos/missing details**: Thanks for the attention to detail! We have updated the draft to address these comments.
>
> **CHI3D**: Thanks for the suggestion! Unfortunately, we were not able to integrate and benchmark on this dataset within the rebuttal time frame.
>
> **Off-the-shelf MAE**: This is the masked-autoencoder pretrained version of the model, we have updated the writing to clarify the unexplained acronym.
>
> **Pose prior**: Thanks for the suggestion! We previously experimented with TokenHMR as a stronger pose prior, but did not observe a noticeable improvement. We find the integration of a generative model an interesting topic that deserves a more thorough investigation. As this goes beyond the scope of the paper, we leave it for future work.

---

> > ### Author Response · Authors · 2024-11-30
> >
> > Dear reviewer VXUb. Please let us know if we have addressed your concerns. If you believe we have, could you please consider raising your score to reflect that?

---

### Official Review · Reviewer_MAjj · 2024-11-04

**Soundness:** 3
**Presentation:** 3
**Contribution:** 3
**Rating:** 8
**Confidence:** 4

**Summary:**

This paper addresses the problem of 3D human pose and shape estimation and cross-frame tracking. The proposed method estimates 3D poses for all the persons in a frame and then updates the tracking poses and bounding boxes. A novel pose update module with GRU is introduced to update the tracking results and store the hidden information not present in the SMPL parameters. The proposed method is benchmarked on both bounding box tracking and pose estimation tasks and demonstrated better performance than the previous method. The comparisons on per-frame runtime also demonstrated its much more efficient design compared to the previous method.

**Strengths:**

1. The proposed method estimates the tracking and pose simultaneously, which could take advantage of prior information on humans.
2. The design choice of hidden state and GRU is sound. Keeping tracking alive seems to be a valid approach to address the missing people in a shorter period.
3. The evaluation results on PoseTrack benchmarks are impressive, validating the proposed method's design choice.

**Weaknesses:**

Overall, the paper is of good quality, but there are still some weaknesses that the reviewer would like to discuss here and in the following questions.
1. The implementation details and model architecture are unclear. The model architecture of both the pose estimation module and the pose update module is unclear. It's also unclear how the detection of bounding boxes is achieved.
2. Although the hidden state and GRU are sound approaches, the experiments do not rigorously evaluate their effectiveness.
3. The proposed model takes advantage of multiple training datasets, although it's not uncommon in the community of human pose estimation, but part of the improvements in the tracking results might be attributed to this.

**Questions:**

1. The videos show unstable pose estimation for static humans, which should be addressed better in the pose tracking method than in the single-frame human pose estimation method. What is the cause of such instability, as the model has information about previous estimations and a learned hidden state?
2. The authors mentioned using a 'strict' threshold in evaluating pose track metrics. Could the authors elaborate more on the choice of threshold and its relationship with the tracking method's precision and recalls?
3. Will the proposed method be open-sourced? There is only limited information on the model architecture and hyper-parameters. It will be hard to reproduce the proposed method if the source code and checkpoint are not available.
4. The proposed method has worse results compared to CLIFF in Table 3. What's the possible cause of it?
5. Dense object tracking has been much improved by papers like CoTracker. How will the proposed method perform when compared to the method Cotracker? It will be interesting to see the performance difference between a 3D-aware object-specific method performed compared to a generic method.

---

> ### Author Response · Authors · 2024-11-28
>
> **Unclear architecture details**: We have added a more thorough description of the architecture in the appendix. We also plan to release inference code for the model and pretrained weights for reproducibility.
>
> **Hidden state**: We conducted an experiment on the effect of the hidden state on performance (Sec. A.1.2). By removing the hidden state we observe noticeably worse performance on challenging PoseTrack scenes and an increase in ID switches by 25%. The effect is smaller on 3DPW validation scenes which may be explained by the fact that most scenes only involve one or two people, where one can assume that most transitions from adjacent frames can be handled trivially without the more sophisticated modeling that the hidden state might provide.
>
> **Multiple datasets**: One of the key motivations of our efforts was to design and build an approach that could be trained on video. Other approaches avoid this by breaking down the problem into sub-problems and training individual modules. We demonstrate the benefits of building a single, learnable stack that can be unrolled across video frames performing the core problems of pose estimation and tracking together. As such, we make use of training data not leveraged by prior work. That said, we would like to highlight that there is substantial overlap in datasets, particularly large scale data like InstaVariety which consists of 3 million images. One of our key video datasets is PoseTrack, and 4D Humans trains on MPII which consists of individual images from the same data source. The main dataset that differs is BEDLAM for which we provide an ablation showing that it is not as critical for maintaining good 2D pose tracking performance on PoseTrack but is important for preserving good 3D estimates unrolled across time (Appendix, Sec A.2.2).
>
> **Unstable pose estimates**: We hypothesize that this behavior could be due to noisy 2D supervision in our video training data. As 2D annotations can be explained by multiple 3D poses, there is limited supervision on 3D pose consistency across frames.
>
> **Strict threshold**: This is the threshold on detection scores above which we accept detections from the bounding box model. A higher threshold reduces the chances of including background figures and smaller or more unusual poses. The detections that are left are generally clearer and easier to follow, hence the improved precision. By default we use a threshold of 0.3. For the strict setting we use 0.7.
>
> **Performance versus CLIFF**: The performance on 3DPW is curious. As a point of comparison, we would like to draw the reviewer’s attention to the respective numbers for CLIFF, HMR 2.0a, and HMR 2.0b on 3DPW. The authors of 4D Humans observed they could train a model that was competitive on 3DPW (HMR 2.0a) but not as good in-the-wild, and with longer training they had a more robust model (HMR 2.0b) that did worse on 3DPW. One possibility to explain this is annotation discrepancies. In-the-wild 2D annotations are fairly different from SMPL annotations, and it is possible that as a result of fitting both during training, we perform worse on 3DPW. In our response to Reviewer VXUb we discuss some investigations we undertook to deal with annotation discrepancy. Occasionally this resulted in models with better 3DPW performance, but we did not pursue this further and instead focused our efforts on in-the-wild numbers. We suspect a large-scale consistently-annotated dataset would go a long way to resolving this issue.
>
> **CoTracker**: Dense point tracking methods like CoTracker are a fascinating line of work, and we are excited about the possibilities they enable. We find however setting up a fair comparison nontrivial. For instance, dense tracking approaches are typically instantiated from pixel positions, whereas many of the body joints that are tracked don’t necessarily correspond to visible surface positions (e.g. the hips). For a fair comparison, all keypoints of the pose would have to be visible in the first frame for the dense tracker, which strongly limits the type of videos for evaluation. One advantage of tracking a pre-defined discrete representation is that many downstream use cases are made easier working directly with an output such as kinematic joint angles which would be difficult to parse from a set of point trajectories.

---

> > ### Comment · Reviewer_MAjj · 2024-11-30
> > **Thanks for the clarification**
> >
> > Thanks for the clarifications from the authors. I am inclined to keep my current rating.

---

### Official Review · Reviewer_ZfVY · 2024-11-04

**Soundness:** 3
**Presentation:** 3
**Contribution:** 2
**Rating:** 6
**Confidence:** 4

**Summary:**

This paper addresses a challenging problem of multi-person 3D pose tracking. Compared to tracking-by-detection pipelines, this paper extends the tracking-by-attention pipelines from Multi-Object Tracking to Multi-person pose tracking, which detect new humans and update existing tracks by learned tokens. Due to the lack of complete training data, this paper explores the mixed datasets with different forms of supervision. The experimental results demonstrate superior performance over other state-of-the-art methods on several benchmarks.

**Strengths:**

1. This paper considers a new pipeline to address multi-person video-based tracking and pose estimation.
2. The paper is well-written and easy to understand. The qualitative results clearly show the advance of the proposed method.
3. This method is very fast compared to other methods and achieve good results in several benchmarks.

**Weaknesses:**

1. This paper is more like a technical report rather than research paper. The contributions are hard to recognise and not clearly highlighted. This makes this paper less insightful.
2. To make the contributions clearer, i would recommend giving some preliminaries about the tracking-by-attention pipelines applied in other applications, e.g., MeMOTR, to distinguish what’s new in your method and what has been done.
3. One important claim in your paper in CoMotion can utilise subtle pixel cues in Line52-53, which is not elaborated is supported in the following text and experiments.
4. The compared methods in tab.1 seems not very recent, what’s the comparison between CoMotion and more advanced MOT methods?
5. Some minor issues:
- Metrics can be introduced. If the space is not enough, you can include them in the appendix.
- I would recommend give short descriptions about ablation studies instead of checking them directly in appendix.

**Questions:**

I really appreciate authors' hard work, which gives fast and accurate results on the benchmarks. However, I am doubtful of this paper's novelty and insight and I give borderline reject accordingly.

---

> ### Author Response · Authors · 2024-11-28
>
> **Contributions & recent methods**: We appreciate the feedback! We believe we are one of the first methods to train an online 3D multi-person pose system that handles crowded in-the-wild scenes with diverse and challenging poses.
> * TRACE (Sun et al.) also predicts multi-person 3D poses in videos, but has not been demonstrated to work on scenes with more than 2-3 people.
> * The only approaches that benchmark on the full target setting of our method are PHALP and its follow up 4D Humans. Both methods take a fundamentally different approach - they predict individual 3D poses on cropped bounding boxes and link them together into tracks. In contrast, CoMotion consists of a unified model that explicitly is trained on video to reason about all people in the scene simultaneously and model their motion through time. In order to achieve this, we push tracking-by-attention into a much more difficult setting modeling a richer signal (3D pose) in camera coordinate frame, training on longer sequences rather than pairs or small handfuls of frames, and modeling through occlusion. This is only possible with a nontrivial combination of datasets and insights around how best to leverage single image and video data with 2D and 3D annotations. The scale and scope of this work unifies several fields (tracking, 3D pose estimation, human motion modeling) and we believe this undertaking and the results are a significant contribution in their own right. We elaborate below on some of the differences to the bounding-box tracking literature.
>
> **Contrast to other tracking-by-attention methods**: CoMotion targets multi-person pose estimation in 3D and tracking jointly. Tackling this task poses several additional challenges which are typically not handled by existing bounding-box tracking-by-attention approaches.
>
> * CoMotion differs from prior tracking-by-attention approaches in its handling of state. MeMOTR in particular does not make predictions for unconfident boxes and leaves occluded queries idle in case they are relevant again. CoMotion in contrast is trained to keep making pose predictions even if the person is occluded or out-of-bounds.
> * A key contribution of our approach is the multi-stage curriculum across a suite of datasets, which starts on single images and ramps up training on longer video sequences. This also allows us to take advantage of the strengths of the various training datasets at our disposal, and considerably improves performance with each stage of the curriculum as demonstrated by the ablation experiment in Sec. A.2.3. We are not aware of any prior work supervising bounding-box trackers that would use a similar curriculum.
> * Making predictions in a 3D coordinate frame requires explicitly accounting for camera intrinsics such that results are projected correctly to the image. This is in contrast to bounding-box trackers that only operate in the image plane.
> * Ground truth data for multi-person 3D pose tracking is scarce. Most datasets only provide a limited set of annotations, which do not provide full supervision for the task or not at the required level of diversity to generalize to in-the-wild scenes. To nevertheless build a system that can operate on complex in-the-wild videos and predict 3D poses of multiple people, we harvest several datasets that contain mutually inconsistent annotations. We added further details in Sec. A.4 of the Appendix.
>
> **Subtle pixel cues**: The qualitative results included in our introductory figure serve as examples of this behavior. Unfortunately, heavily occluded figures are seldomly annotated in benchmark datasets, which prevents adequately quantifying the phenomenon.
> **Metrics and ablation discussion**: We have updated the paper with more details, thanks! We have added brief explanations of the metrics as well as an additional section in the main paper summarizing the results of our controlled experiments.

---

> > ### Author Response · Authors · 2024-11-30
> >
> > Dear reviewer ZfVY. Please let us know if we have addressed your concerns. If you believe we have, could you please consider raising your score to reflect that?

---

> > > ### Comment · Reviewer_ZfVY · 2024-12-03
> > > **Thank you for your response**
> > >
> > > Thank you for your explanations and clarifications. I would suggest adding the comparisons against other tracking-by-attention methods in the final revision. I raise the score to. 6.

---

### Author Response · Authors · 2024-11-28

We thank all reviewers for their valuable feedback and respond to concerns and questions directly after each review.

---

### Meta-Review · Area_Chair_4yqd · 2024-12-23

**Metareview:**

## Summary
This paper presents a multi-person 3D pose tracking method that extends tracking-by-attention pipelines, detecting new humans and updating existing tracks using learned tokens. The method uses mixed datasets and different supervision forms, demonstrating superior performance in various benchmarks. The method also introduces a novel pose update module with GRU, improving performance and efficiency in bounding box tracking and pose estimation tasks.

## Strengths
* The method solves a relevant problem of tracking multiple people from a monocular camera; tracking-by-attension method in crowded scenes, leveraging prior human information.
* 3D human pose update module design is effective and fast and achieves good results in several benchmarks.
* Design choice of hidden state and GRU is sound.

## Weaknesses
* The tracking experiment is primarily conducted on the PoseTrack dataset.
* Preliminaries about tracking-by-attention pipelines applied in other applications, such as MeMOTR, could help distinguish new contributions.
* The comparison between CoMotion and more advanced MOT methods is not very recent.
* The proposed model takes advantage of multiple training datasets, which could contribute to improvements in tracking results.
* The authors mentioned using a'strict' threshold in evaluating pose track metrics, which could be elaborated on.
* The proposed method has worse results compared to CLIFF, which could be due to various factors.
* The authors do not add tracks during training, which could potentially be detrimental to learning.
* The authors do not expect the shape of the mesh to correspond to the person’s physical body.
* The authors should provide more details on how they apply the regularization for the beta optimization.
* The authors could utilize the CHI3D [a] as an additional benchmark due to its well-tracked 3D poses and more challenging human-human interactions.

##Conclusion
Although there are a few weaknesses, the paper tackles an interesting problem and proposes a well defined solution evaluated on several benchmarks. Based on the reviews and author's feedback, the paper should be accepted but should include all the points raised by the reviewers.

**Additional Comments On Reviewer Discussion:**

There were two reviewers that discussed with the authors, keeping the same score and even increasing. Therefore, most of the concerns were satisfied by the author's feedback.

---

### Decision · Program_Chairs · 2025-01-22

Accept (Poster)